# Predictive Value of Dual-Energy CT-Derived Metrics for the Use of Bone Substitutes in Distal Radius Fracture Surgery

**DOI:** 10.3390/diagnostics14070697

**Published:** 2024-03-26

**Authors:** Philipp Reschke, Vitali Koch, Scherwin Mahmoudi, Christian Booz, Ibrahim Yel, Jennifer Gotta, Adrian Stahl, Robin Reschke, Jan-Erik Scholtz, Simon S. Martin, Tatjana Gruber-Rouh, Katrin Eichler, Thomas J. Vogl, Leon D. Gruenewald

**Affiliations:** 1Department of Diagnostic and Interventional Radiology, Clinic for Radiology and Nuclear Medicine, Hospital of the Goethe University Frankfurt, 60590 Frankfurt am Main, Germany; philipp.reschke@outlook.de (P.R.); vitali-koch@gmx.de (V.K.); dr.ibrahimyel@gmail.com (I.Y.); jennifergotta@aol.com (J.G.); s1678625@stud.uni-frankfurt.de (A.S.); gruenewald.leon@me.com (L.D.G.); 2Department of Dermatology and Venereology, University Medical Center Hamburg-Eppendorf, 20537 Hamburg, Germany

**Keywords:** bone density, bone substitutes, osteoporosis, osteoporotic fractures, bone diseases, computed tomography

## Abstract

(1) Background: Low bone mineral density (BMD) is a significant risk factor for complicated surgery and leads to the increased use of bone substitutes in patients with distal radius fractures (DRFs). No accepted model has yet been established to predict the use of bone substitutes to facilitate preoperative planning. (2) Methods: Unenhanced dual-energy CT (DECT) images of DRFs were retrospectively acquired between March 2016 and September 2020 using the internal PACS system. Available follow-up imaging and medical health records were reviewed to determine the use of bone substitutes. DECT-based BMD, trabecular Hounsfield units (HU), cortical HU, and cortical thickness ratio were measured in non-fractured segments of the distal radius. Diagnostic accuracy parameters were calculated for all metrics using receiver-operating characteristic (ROC) curves and associations of all metrics with the use of bone substitutes were evaluated using logistic regression models. (3) The final study population comprised 262 patients (median age 55 years [IQR 43–67 years]; 159 females, 103 males). According to logistic regression analysis, DECT-based BMD was the only metric significantly associated with the use of bone substitutes (odds ratio 0.96, *p* = 0.003). However, no significant associations were found for cortical HU (*p* = 0.06), trabecular HU (*p* = 0.33), or cortical thickness ratio (*p* = 0.21). ROC-curve analysis revealed that a combined model of all four metrics had the highest diagnostic accuracy with an area under the curve (AUC) of 0.76. (4) Conclusions: DECT-based BMD measurements performed better than HU-based measurements and cortical thickness ratio. The diagnostic performance of all four metrics combined was superior to that of the individual parameters.

## 1. Introduction

Distal radius fractures (DRFs) occur frequently in elderly patients, constituting approximately 18% of all fractures [1]. Osteoporosis is a known risk factor for fragility fractures. In particular, females over 65 years of age are more susceptible to DRFs due to the high incidence of osteoporosis in this age group [2]. 

With the introduction of volar locking plates, the surgical intervention rate for DRFs has increased. A volar locking plate combined with bone augmentation for osteoporotic bone fractures provides better biomechanical fixation [3]. Typically, the decision to use bone substitutes for bone defects is made during surgery based on the texture of the bone. Using bone augmentation in DRRs can result in a longer operation time and carries a risk of complications such as intra-articular escape of bone substitutes and infection. Preoperative knowledge of the likelihood of using bone substitutes can help surgeons minimize these risks by selecting appropriate materials, determining their placement, and ensuring their containment within the intended area. Furthermore, it enables the patient to be informed beforehand about the procedure details. This foresight could play a crucial role in reducing the duration of the surgery and in preparing for any complications that might arise [4,5,6]. 

Multiplanar CT is the method of choice for dedicated preoperative planning. Quantitative CT (QCT) offers a precise method for measuring the volumetric bone mineral density in trabecular bone. For calibration, a phantom is conventionally placed below the patient’s body [7]. Additionally, in-scan calibration prevents a retrospective application. In response to this limitation, alternative indicators for bone density, including HU-based measurements and cortical thickness ratio, have been suggested [8,9,10,11]. However, these methods are subject to various limitations. The variability in the distribution and composition of bone substance and marrow leads to inaccuracies in HU-based measurements [12]. Dual-energy CT (DECT) employs varying X-ray spectra, enabling the differentiation of materials. Using a postprocessing algorithm that employs DECT, bone mineral density (BMD) can be volumetrically assessed without the need for a physical phantom. Routine examinations have validated the algorithm’s applicability [13,14,15,16]. 

Previously, it has been demonstrated that this algorithm can predict the use of bone substitutes in DRF patients [17]. However, DECT scanners are not widely available due to high acquisition costs [18]. Our objective was to assess whether opportunistic bone mineral density assessments could serve as a reliable alternative for predicting the use of bone substitutes. We hypothesized that the diagnostic accuracy of DECT-based BMD assessment for predicting the use of bone substitutes in DRF patients is greater than that of opportunistic bone mineral assessments.

## 2. Materials and Methods

The institutional review board approved this retrospective study and waived the requirement to obtain written informed consent. Our study constitutes a secondary analysis of pre-existing study data. In a different study context, the study data were partially previously published by Gruenewald et al. and Reschke et al. [16,17].

### 2.1. Patient Selection

CT scans of adult patients who underwent DECT scans of the distal radius between March 2016 and September 2020 were retrospectively obtained at the University Hospital Frankfurt through the internal picture archiving and communication (PACS) system. The exclusion criteria were severe destruction of the distal radius, suspected or confirmed malignancies, metallic implants, osteomalacia, and unavailability of medical records. 

### 2.2. Imaging Protocol

Non-contrast CT images of the radius were acquired by a third-generation dual-source CT device (SOMATOM Force; Siemens Healthineers, Erlangen, Germany) in dual-energy mode. The X-ray tubes were operated at different kilovolt (kV) settings (tube 1: 90 kVp, 180 mAs; tube 2: Sn150 kVp 8 [0.64 mm tin filter], 180 mAs). The system utilized automatic tube current modulation (CARE dose 4D; Siemens Healthineers). Multiplanar CT images were reconstructed with a specialized dual-energy bone kernel (Br69f). The thickness of the image slices was 1 mm with an increment of 0.75 mm. 

### 2.3. Image Interpretation

Two radiologists, with fourteen years of experience and six years of experience in musculoskeletal imaging, independently performed HU-based bone density assessments and cortical thickness ratio measurements using preoperative CT images of DRFs. Both radiologists were unaware of the patients’ clinical symptoms and injury mechanisms. In divergent assessments, a third radiologist with ten years of experience in musculoskeletal imaging was consulted to establish a majority decision. Surgical reports and follow-up imaging were reviewed for the use of bone substitutes. Cancellous bone autografts from the iliac crest or the iliac spine and hydroxyapatite ceramics (RESORBA Medical, Nuremberg, Germany) were used for bone substitution. 

### 2.4. BMD Assessment

Volumetric BMD assessment was carried out manually by a single board-certified radiologist with seven years of experience in musculoskeletal imaging. Delineation of non-fractured segments of the DRF was performed as previously reported [17]. DECT image series were used as inputs for phantomless volumetric BMD evaluation with specialized software (BMD Analysis, Fraunhofer IGD, Darmstadt, Germany, version 5.0.1). This software employs a material decomposition algorithm that distinguishes between the five components of trabecular bone: water, calcium hydroxyapatite, collagen matrix, red marrow, and adipose tissue for each voxel, as previously described [14]. In brief, the volume of the collagen matrix and bone mineral, as well as the volume of water and red marrow, were assessed together. Solving the resulting three-material decomposition involves introducing an additional condition in which the sum of the remaining three fractional volumes equals 100% in each voxel. We standardized the assessment methods in terms of location, area, and volume. First, the length of the line from the styloid process of the radius to the distal radioulnar joint was measured. A subsequent line, half the length of the line, was drawn proximally along the medial edge of the radius. The terminal point of the line served as the reference for drawing a rectangle with a height equivalent to that of the line, covering the distal radius centered at the end point of the line. The most extensive intact segment within the specified rectangle was carefully outlined across all 2D slices to create a three-dimensional region of interest (ROI) for each patient. If all intact segments in the rectangle covered less than 25% of the rectangle’s volume, the scan was deemed inadequate for BMD assessment, leading to the exclusion of the patient from the analysis. Within the designated rectangle, the trabecular bone region was outlined across all 2D slices to form a three-dimensional region of interest (ROI) for the evaluation of DECT-based BMD. Next, the three-dimensional ROI, alongside the DECT image series, was input into a secondary software tool (BMD Analysis; Fraunhofer IGD), which employs a specialized material decomposition algorithm to calculate the BMD. Trabecular HU measurements were derived from the same region of interest (ROI) used for BMD assessment. The average of the trabecular HU of all 2D slices within the rectangle was calculated. Additionally, cortical HU values were derived by manually measuring the HU of the anterior and posterior cortical bone at the center of the rectangle and calculating their average value. The cortical thickness ratio was determined by dividing the external diameter of the radius by its internal diameter. Additionally, the circles outlined for measuring cortical HU values were used as reference points for the anterior and posterior cortical bone when measuring the cortical thickness ratio (Figure 1).

### 2.5. Statistical Analysis

The statistical analysis was conducted using MedCalc (Windows Version 20.1, MedCalc, New York, NY, USA) and R (Windows Version 4.2.2, the R Foundation, Vienna, Austria). To assess normal distribution, the Kolmogorov–Smirnov test was applied. Differences in baseline characteristics were assessed using unpaired *t*-tests, the Mann–Whitney test for continuous variables, and Fisher’s exact test for categorical variables. Age, biological sex, bone mineral density (BMD), trabecular HU, cortical HU, and cortical thickness ratio are presented as medians with interquartile ranges in parentheses. Comparative analysis of the performance of various bone density measurements in predicting bone substitutes was carried out through receiver operating characteristic (ROC) curves and precision–recall (PR) curves. Pairwise comparisons of ROC curves were performed using the DeLong method. The partial area under the curve (AUC) with a false positive rate (FPR) of 0.2–0.4 was evaluated using bootstrap analysis. We defined an FPR above 0.4 as unacceptable for partial AUC to avoid pretest scores that produce excessive false positives. The Youden Index was used to determine the optimal threshold for predicting the use of bone substitutes. Regression analysis was performed using a multivariable logistic regression model adjusted for age and female sex to assess associations of DECT-based BMD, cortical HU, trabecular HU, and cortical thickness ratio with the use of a bone substitute. First, univariate logistical regression was performed for each indicator of bone density separately to compare the performance of the different indicators against each other. Subsequently, multivariate logistical regression was performed using all the indicators. Statistical significance was defined at a threshold of *p* < 0.05.

## 3. Results

### 3.1. Patient Characteristics

A total of 321 patients were considered for study inclusion; 14 were excluded due to metallic implants in the distal radius, 12 were excluded due to known or suspected malignancy and 11 were excluded because the health records were unavailable. No patient was excluded because of osteomalacia. A total of 22 patients were excluded due to severe post-traumatic destruction of the radius spanning > 75% of the target area. Thus, the study population comprised 262 DRF patients (median age 55 years [IQR 43–67 years]; 159 females, 103 males) (Figure 2). 

Of the 262 patients in our study, 28 patients received bone substitutes. Patients with DRF who received bone substitutes were slightly older than those who did not (median age 60 years [IQR 52–66 years] vs. 54 years [IQR 40–67 years]). There was no significant difference in fracture severity between patients who received bone substitutes and those who did not (*p* = 0.97). C fracture was the most common fracture type for the patients who received bone substitutes (79% of all fractures in this group) and those who did not (74% of all fractures in this group) (Table 1).

DECT-based BMD assessments of patients who received bone substitutes were significantly lower than those of patients who did not receive bone substitutes (median BMD 79.9 mg/cm^3^ [IQR 69.6–89.9 mg/cm^3^] vs. 93.4 mg/cm^3^ [IQR, 78.2–109.4 mg/cm^3^], *p* < 0.001). (Figure 2). In contrast, there were no significant differences in the trabecular HU-or cortical-HU values between DRF patients who received bone substitute and those who did not (median trabecular HU, 22.5 HU [IQR −0.25–47 HU] vs. 35 HU [IQR −19.5–89 HU] *p* > 0.05; median cortical HU, 1597.5 HU [IQR 1447–1712.5 HU] vs. 1713 HU [IQR 1550.75–1845.25 HU] *p* = 0.03). The cortical thickness ratio was slightly lower for patients who received bone substitutes than for those who did not (1.34 [IQR 1.27–1.42] vs. 1.37 [IQR 1.29–1.45]) (Table 1, Figure 3).

As bone density tends to decrease with age, there is a strong negative correlation between age and DECT-based BMD (Figure 4, Pearson coefficient, DECT based BMD: −0.7). None of the other metrics correlated with age as strongly as DECT-based BMD did (Pearson coefficient of cortical HU: −0.16, trabecular HU: −0.27, cortical thickness ratio: −0.08). We observed a significant difference between males and females in DECT-based BMD (*p* < 0.001), trabecular HU (*p* < 0.01) and cortical thickness ratio (*p* = 0.03), with DECT-based BMD and trabecular HU showing the most substantial differences. Conversely, cortical HU did not exhibit a significant difference between the sexes.

### 3.2. Logistic Regression Analysis

After adjustment for age and sex, increased DECT-based BMD was associated with lower odds of using bone substitutes (BMD odds ratio, 0.95, *p* < 0.05). The model remained significant after adjustment for female sex (*p* < 0.05). Logistic regression models for the other bone density assessment methods did not yield significant results (cortical HU odds ratio 0.99, *p* = 0.06; trabecular HU 0.99, *p* = 0.33; cortical thickness ratio odds ratio 0.04, *p* = 0.21) (Table 2).

### 3.3. Optimal Threshold 

An optimal DECT-based BMD cut-off value of 97.9 mg/cm^3^ yielded a sensitivity of 90% and a specificity of 47%. A cortical HU cut-off value of 1778 HU resulted in the same sensitivity of 90% and a lower specificity of 40%. The optimal thresholds for trabecular HU and cortical thickness ratio generated lower specificity and sensitivity (Table 3, Figure 4). 

### 3.4. Comparative Analysis of Measurement Methods via ROC Curves

A pairwise analysis of ROC curves revealed that the AUC of BMD was significantly greater than the AUC of trabecular HU (*p* < 0.05), but not for the other metrics. A false positive rate of less than 0.2 leads to a true positive rate of less than 0.5 for all the metrics in our study. The partial AUC (FPR AUC 0.2–0.4) of DECT-based BMD was significantly higher than the AUC of trabecular HU and cortical thickness but not significantly different from the AUC of cortical HU. The AUC of our combined model of all metrics was significantly greater than the AUC for trabecular HU and cortical thickness. The partial AUC of the combined model (FPR AUC 0.2–0.4) was significantly higher than all other metrics (Table 4, Figure 5).

## 4. Discussion

Previous studies have demonstrated the effectiveness of DECT in predicting the need for bone substitutes in patients with DRFs [17]. 

This study addresses a notable gap in the literature by comparing various predictive parameters for the use of bone substitutes. Our study population is representative in terms of age and sex (median age 55 years [IQR 43–67 years]; 159 females, 103 males). Generally, distal radius fractures are common in women older than 50 years, due to osteoporosis [19].

We assessed the diagnostic accuracy of DECT-based BMD assessment compared to that of HU-based metrics and cortical thickness ratio in predicting the use of bone substitutes in patients with DRF. As a result, DECT-based BMD outperformed HU-based metrics and cortical thickness ratio. In contrast to HU-based metrics and cortical thickness ratio, the DECT-based BMD assessment yielded a statistically significant prediction model for bone substitution (*p* < 0.05), and increased DECT-based BMD was significantly associated with lower odds of using bone substitutes. 

A potential reason is that DECT-based material decomposition allows precise tissue analysis, to differentiate mineralized bone from surrounding tissue. There are several limitations to opportunistic bone mineral density assessments. The cortical thickness ratio relies on a simplified two-dimensional measurement for evaluating the three-dimensional structure of bones. Past research has shown that variations in bone marrow and body composition can lead to inaccuracies in HU-based bone density assessments [12,20]. 

Our combined model’s partial AUC was significantly greater than that of all the other metrics. The combined model of all four metrics could be reported together with a risk assessment stratification for needing bone substitutes to the on-call surgeon. 

Previous studies have revealed that delaying internal fixation 24–92 h after injury can stimulate fracture callus formation compared to immediate internal fixation. We suggest that proper timing of internal fixation for patients at risk of bone defects could contribute to avoiding the unnecessary use of bone substitutes [21]. In addition, identifying patients who are at risk of bone defects and complex surgical procedures could necessitate a tightly coordinated follow-up assessment plan to avoid complications. 

Despite the clinically relevant findings, it is important to acknowledge the limitations of our study. First, the cohort largely consisted of patients who underwent CT imaging of the radius after initial X-ray imaging indicated a complex fracture. Consequently, individuals with straightforward, uncomplicated fractures may not be adequately represented in our study. Second, approximately 18% of our study cohort was excluded according to predefined criteria, which may introduce selection bias to our study. Third, the study was conducted at a single institution, which may limit the generalizability of the findings. The predictive models developed should be externally validated in further studies. Fourth, the algorithm employed in this research may not be universally accessible or compatible with diverse hospital systems. Fifth, the applicability of the study’s findings is constrained by the availability of DECT scanners. Sixth, different healthcare facilities may employ different acquisition protocols for DECT. In our study, we adhered to a fixed DECT-acquisition protocol. This standardization aimed to ensure reliable and consistent BMD measurements. Seventh, we must consider the challenges related to the combined model’s feasibility and applicability in clinical workflows.

We identified the delineation process for DECT-based BMD and trabecular HU as the most time-consuming step. Future research should focus on automatic delineation, inter-device reproducibility and clinical workflow integration of the different predictive parameters.

## 5. Conclusions

We demonstrated that DECT-based volumetric BMD assessment is a streamlined tool for risk stratification for bone substitutes and is the sole metric significantly associated with the use of bone substitutes. Furthermore, the combined model of all parameters provides a more detailed risk assessment for the use of bone substitutes. 

## Figures and Tables

**Figure 1 diagnostics-14-00697-f001:**
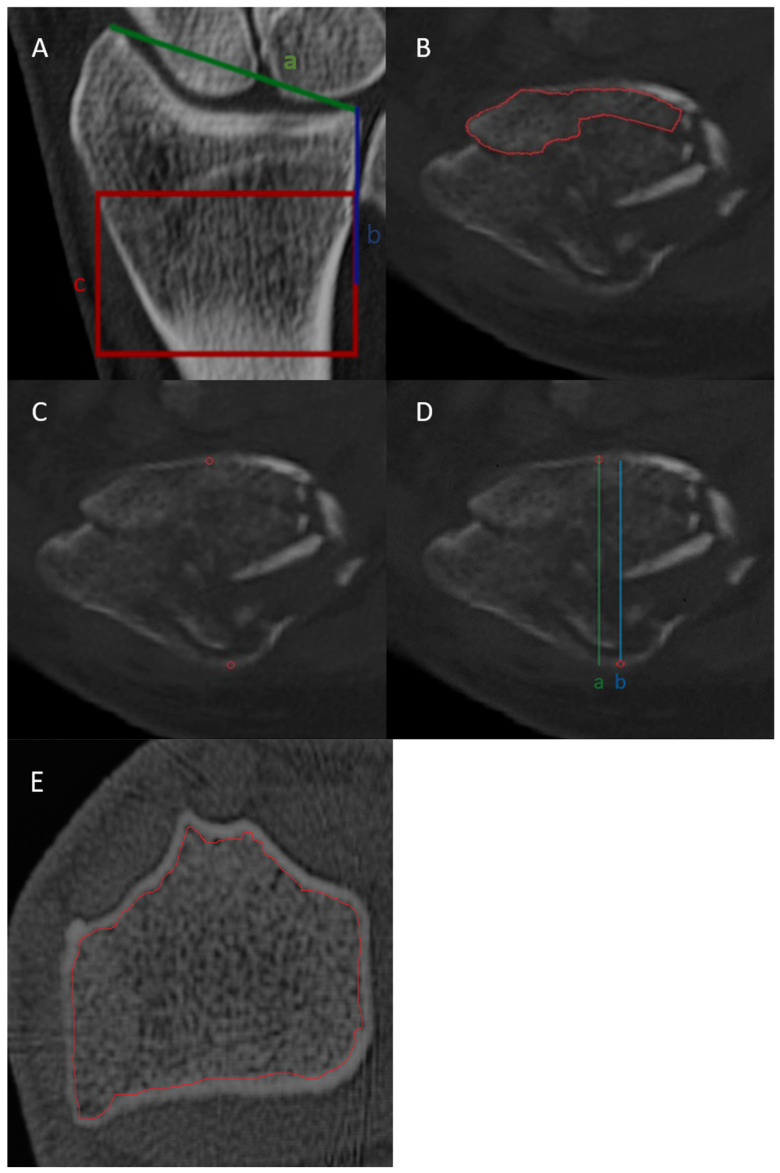
Definition of the metaphyseal area of interest. (**A**) To normalize the location for opportunistic BMD assessment, a line connecting the styloid process and the distal radioulnar joint was drawn (a). A second line (b) measuring half the length of line (a) was then drawn proximally along the medial border of the radius (b). The end of line (b) was used to construct a rectangle (c) with the height of (b). (**B**) Within the rectangle, the trabecular bone was delineated throughout the entire stack of 2D slices to obtain a three-dimensional region of interest (ROI) for DECT-based BMD assessment and trabecular HU measurement. In this example, the intact segment covers less than 25% of the rectangle’s volume. If all intact segments in the entire stack of 2D slices covered less than 25% of the rectangle’s volume, the patient was excluded from the analysis. (**C**) Cortical HU values were obtained by manual measurement of the anterior and posterior corticalis (circles in red) and by calculation of the mean value. (**D**) The cortical thickness ratio was obtained by dividing the outside diameter (a) of the radius by the inside diameter (b). The circles outlined for measuring cortical HU values were used as reference points for the anterior and posterior cortical bone (**E**) The non-fractured segment in the rectangle covers the whole volume within the rectangle in this 2D slice, and the patient was included in the study.

**Figure 2 diagnostics-14-00697-f002:**
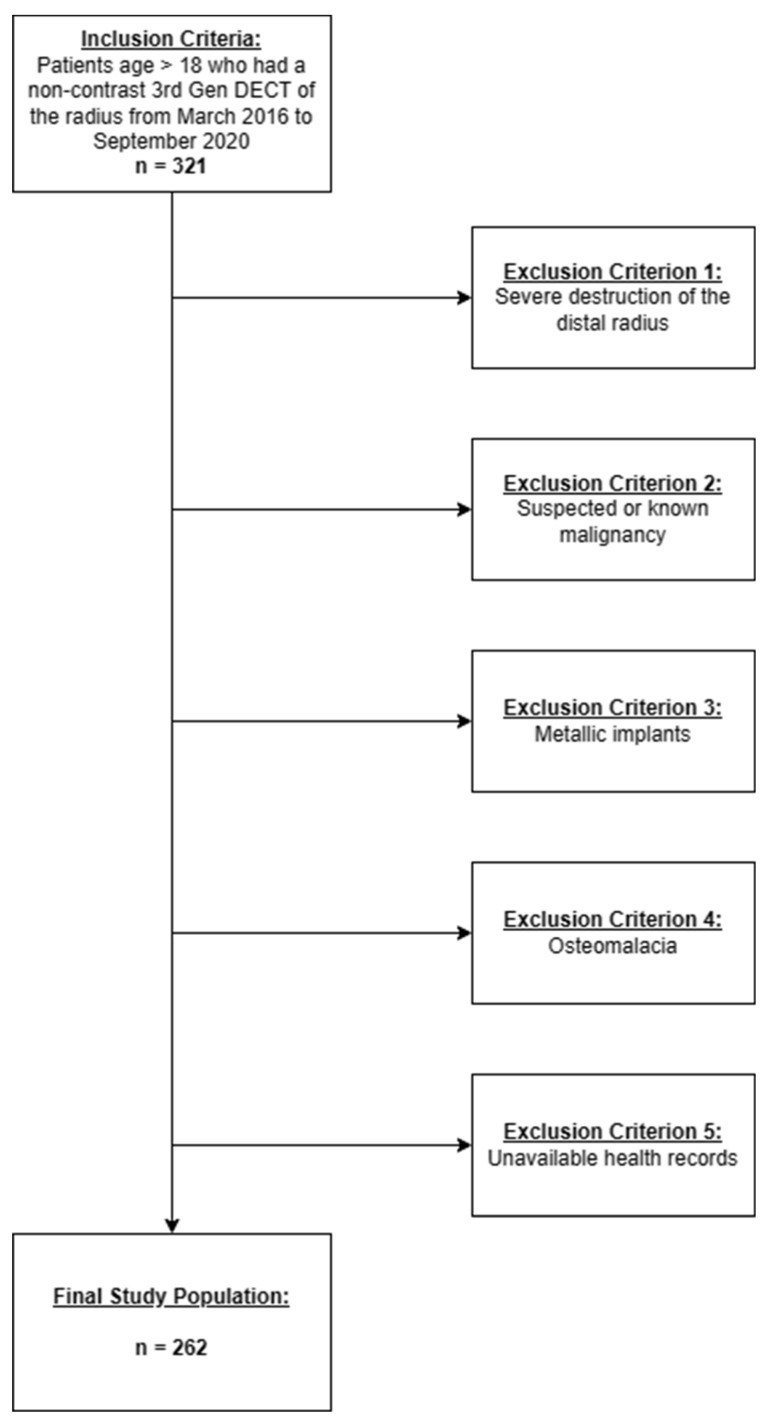
Flow chart of patient inclusion and exclusion criteria and standards of reporting.

**Figure 3 diagnostics-14-00697-f003:**
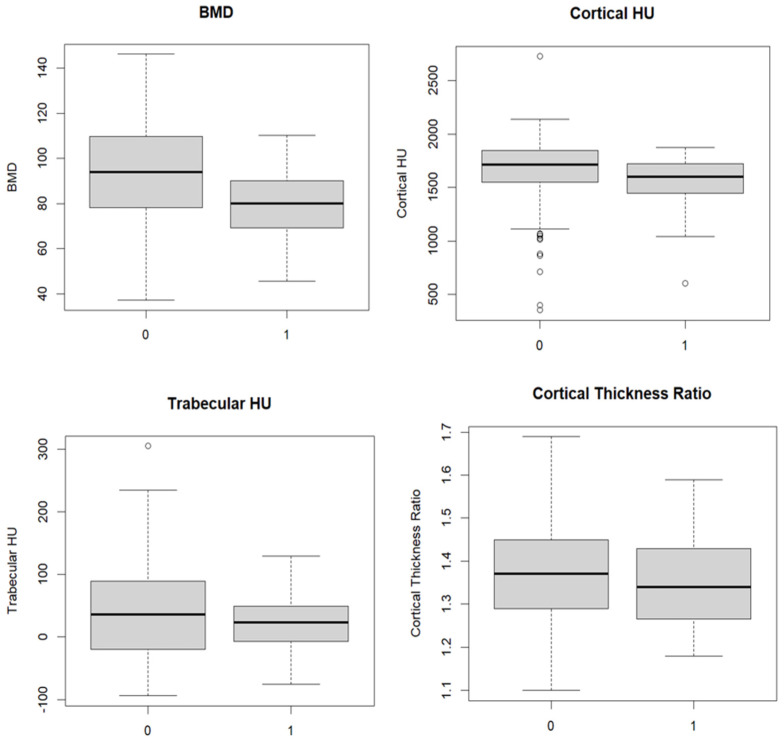
Box plots showing the distribution of values of the different bone density measurement methods with bone substitutes (1) and without bone substitutes (0).

**Figure 4 diagnostics-14-00697-f004:**
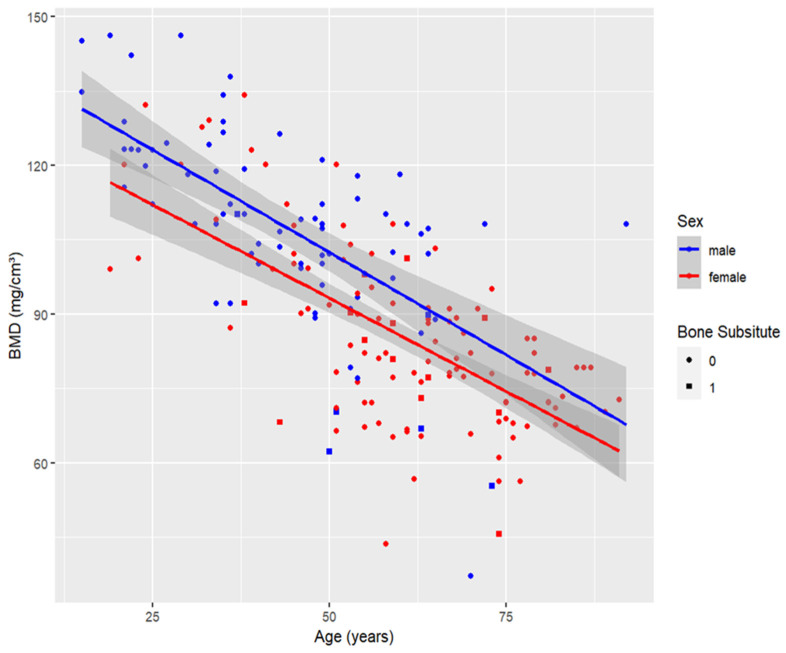
Scatter plot showing volumetric bone mineral density (BMD) values with and without bone substitutes, by age in years, of the total study population (*n* = 262). The dark gray area marks the 95% confidence interval.

**Figure 5 diagnostics-14-00697-f005:**
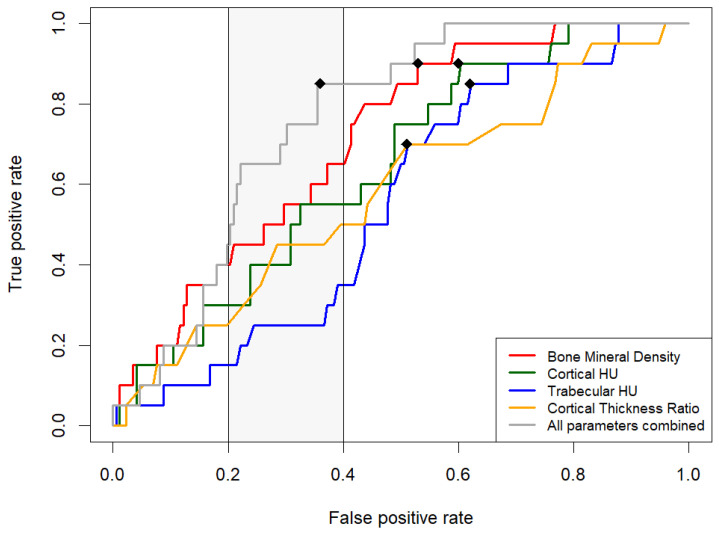
ROC analysis of the performance of all metrics in predicting the use of bone substitutes. The optimal thresholds (maximal Youden Index) are marked by dots. The gray area marks the partial AUC area.

**Table 1 diagnostics-14-00697-t001:** Patient characteristics stratified by the use of bone substitutes after DRF.

Variables	Total DRF (*n* = 262)	No Bone Substitutes (*n* = 234)	Bone Substitutes (*n* = 28)	*p*-Value
Age (years)	55 (43–67)	54 (40–64)	60 (52–66)	0.15
Male (*n*)	103	94	9	
Female (*n*)	159	140	19	0.53
BMD (mg/cm^3^) median (IQR)	91.15 (77.98–108.2)	93.75 (78.18–109.43)	79.85 (69.63–89.9)	0.002
Trabecular HU median (IQR)	31 (−19.5, –84.75)	35 (−19.5–89)	22.5 (−0.25–47)	0.03
Cortical HU median (IQR)	1688 (1531.5–1837)	1713 (1550.75–1845.25)	1597.5 (1447–1712.5)	0.43
Cortical thickness ratio median (IQR)	1.36 (1.29–1.45)	1.37 (1.29–1.45)	1.34 (1.27–1.42)	0.23
A Fracture (*n*)	27	24	3	0.9
B Fracture (*n*)	39	36	3	
C Fracture (*n*)	196	174	22	

The data are expressed as median with interquartile ranges in parentheses. BMD: bone mineral density, DRF: distal radius fracture, HU: Hounsfield Unit, IQR: interquartile range.

**Table 2 diagnostics-14-00697-t002:** Multivariate logistic regression analysis for the use of bone substitutes in DRFs.

Bone Substitutes	Coefficient (β)	Odds Ratio	*p*-Value	*n*
BMD	−0.04	0.96	0.003	262
Age 50–64 (*n* = 15)	−0.03	0.972	0.27	89
Age 65–79 (*n* = 8)	−0.07	0.93	0.095	59
Age ≥ 80 (*n* = 1)	−0.113	0.893	0.592	15
Female Sex (*n* = 19)	−0.02	0.98	0.04	159
Cortical HU	−0.001	0.99	0.06	262
Age 50–64 (*n* = 15)	−0.003	0.99	0.02	81
Age 65–79 (*n* = 8)	−0.003	0.99	0.13	50
Age ≥ 80 (*n* = 1)	0.001	1.001	0.863	15
Female Sex (*n* = 19)	−0.001	0.99	0.17	159
Trabecular HU	−0.003	0.99	0.33	262
Age 50–64 (*n* = 15)	−0.002	0.99	0.02	81
Age 65–79 (*n* = 8)	0.008	1008	0.3	50
Age ≥ 80 (*n* = 1)	−0.5	0.61	0.998	15
Female Sex (*n* = 19)	−0.0003	1.0	0.94	159
Cortical thickness ratio	−2.72	0.07	0.21	263
Age 50–64 (*n* = 15)	−6.42	0.0016	0.07	81
Age 65–79 (*n* = 8)	−1.03	0.036	0.84	50
Age ≥ 80 (*n* = 1)	3.439	3.02	0.999	15
Female Sex (*n* = 19)	−3.28	0.04	0.22	159

Regression analysis was performed using a multivariable logistic regression model to determine the association between BMD and the requirement for bone substitutes. The regression model was adjusted for age and gender. Age-adjusted analyses were conducted using age group stratification to account for potential non-linear associations, with patients younger than 50 years serving as the reference group.

**Table 3 diagnostics-14-00697-t003:** Diagnostic accuracy testing of different predictive values for the use of bone substitutes in DRFs.

Variable	AUC [DeLong]	Partial AUC	Optimal Threshold	TPR	FPR	Specificity
BMD mg/cm^3^	0.71 (0.61–0.81)	0.66	97.9	0.9	0.53	0.47
Cortical HU	0.65 (0.53–0.76)	0.60	1778	0.9	0.6	0.4
Trabecular HU	0.55 (0.44–0.67)	NA	55	0.85	0.62	0.38
Cortical Thickness Ratio	0.58 (0.45–0.72)	0.58	1.37	0.7	0.51	0.49
All variables combined	0.76 (0.68–0.84)	0.79	-	0.85	0.36	0.64

The 95% confidence intervals of the AUC values are shown in parentheses. The partial AUC was calculated for FPR between 0.2 and 0.4. The optimal threshold equals the maximal Youden Index. BMD: bone mineral density, AUC: area under the curve, TPR: true positive rate (sensitivity), FPR: false positive rate (1–specificity), NA: not applicable (curve below diagonal).

**Table 4 diagnostics-14-00697-t004:** Pairwise comparison of ROC curves.

Paired Variables	*p*-Value for AUC	*p*-Value for Partial AUC
BMD vs. Cortical HU	0.43	0.44
BMD vs. Trabecular HU	0.04	0.05
BMD vs. Cortical Thickness Ratio	0.16	0.44
Combined vs. Cortical HU	0.06	0.04
Combined vs. Trabecular HU	0.003	0.001
Combined vs. Cortical Thickness Ratio	0.03	0.03
Combined vs. BMD	0.16	0.02

*p*-value for AUC difference (whole curve) [DeLong], *p*-value for partial AUC difference (FPR 0.2~0.4) [bootstrap]. BMD: bone mineral density, AUC: area under the curve, HU: Hounsfield Unit, FPR: false positive rate.

## Data Availability

Data are contained within the article.

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
