# Peer review of "Predictive Value of Dual-Energy CT-Derived Metrics for the Use of Bone Substitutes in Distal Radius Fracture Surgery"

_diagnostics, 2024, doi:10.3390/diagnostics14070697_

Round 1
Reviewer 1 Report
Comments and Suggestions for Authors
Thank you for sending your manuscript for our consideration. Although the topic is not very novel but is well written and practical.
My only suggestion is to include DECT image(s) and illustrations about how cortical and trabecular HUs and also cortical thickness ratio are measured and calculated.
Reviewer 2 Report
Comments and Suggestions for Authors
Dear authors,
I read with interest the original article entitled „Predictive value of dual-energy CT-derived metrics for the use of bone substitutes in distal radius fracture surgery”. I have the following statements regarding the strengths and weaknesses (limitations) of this study:
Strengths:
- The study uses a relatively large sample size (262 patients), which strengthens the generalizability of the findings.
- It explores multiple parameters for predicting bone substitute use, providing a more comprehensive approach.
- The combined model using all four metrics (BMD, cortical HU, trabecular HU, cortical thickness ratio) shows promise for improved accuracy.
Weaknesses (limitations):
- Hypothesis and aim of the study: Please include a clear definition of the hypothesis and the objective of the investigation.
- Materials and methods: 2.1. Patient selection – line 69, please complete „XXX”.
- Figure 1 – flow chart: For each exclusion criteria mentioned, please provide the number of patients (n =).
- Figure 2 – Scatter plot: Is there a correlation that has been identified? Are correlation coefficients, such as the Pearson correlation coefficient or rank correlation, analysed in your analysis?
- Retrospective design: Since the data was collected after the fact, it can't determine cause and effect. Other factors besides bone density may have influenced the use of bone substitutes.
- Limited generalizability: The study population may not be representative of the entire population with distal radius fractures (age range, specific fracture types).
- AUC of 0.76: While the combined model shows better accuracy than individual parameters, 0.76 on the AUC scale is only considered "fair" discrimination. There could be a significant number of false positives or negatives.
- Lack of cost-benefit analysis: The study doesn't explore the cost implications of using a more complex multi-parameter model compared to the simpler BMD measurement.
Further considerations:
- The study doesn't mention other factors that might influence bone substitute use, such as surgeon preference or fracture severity. Can you provide some details regarding this subject?
- The authors could explore if the combined model performs better in specific patient sub-groups.
- Please include a paragraph with future direction after limitations.
Overall, the study offers some valuable insights into using DECT scans for predicting bone substitute use in distal radius fractures. However, the limitations and the relatively low accuracy of the combined model highlights the need for further research in this area. All the limitations and further considerations should be discussed by the authors.
Comments on the Quality of English LanguageModerate editing of English language is required. Kindly review the text once again and rectify any errors.
Round 2
Reviewer 2 Report
Comments and Suggestions for Authors
Dear Author,
I carefully examined the revised version of the manuscript, entitled "Predictive value of dual-energy CT-derived metrics for the use of bone substitutes in distal radius fracture surgery" with the main objective of assessing whether opportunistic bone mineral density assessments could serve as a reliable alternative for predicting the use of bone substitutes.
I greatly appreciate the changes that were made in accordance with the reviewers' requirements.
A minor revision of the English language is necessary.
Comments on the Quality of English LanguageA minor revision of the English language is necessary.